# Planning Domain Simulation: An Interactive System for Plan Visualisation

**Primary Keywords:** *Applications; Knowledge Representation/Engineering*

## Abstract

Representing and manipulating domain knowledge is essential for developing systems that can visualize plans. This paper presents a novel plan visualisation system called Planning Domain Simulation (PDSim) that employs knowledge representation and manipulation techniques to support the plan visualization process. PDSim can use PDDL or the Unified Planning Library Python representation as the underlying language for modelling planning problems and provides an interface for users to manipulate this representation through interaction with the Unity game engine and a set of planners. The system's features include visualising plan components, and their relationships, identifying plan conflicts, and examples applied to real-world problems. A user evaluation has been conducted to compare PDSim against the standard way using text editors and planners and to evaluate the perceived usefulness and ease of use of PDSim as an additional tool used by students for knowledge representation modelling and automated planning. The benefits and limitations of PDSim are also discussed, highlighting future research directions in the area.

## Introduction

Modelling planning domains that are both correct and robust can be a challenging problem, especially in real-world domains. For instance, consider the following robot planning task: a set of robots are deployed in a factory to help with warehouse logistics. The robots can navigate on a predefined grid map with simple 4-way movements, pick up and drop boxes, and deliver objects to a van parked in the warehouse. The problem also imposes certain limitations: the robots cannot cross each other and the vans can only accept a specific box. The above problem could be viewed as a slightly modified version of the sequential *Floor Tile* domain from the 2011 International Planning Competition (IPC):[1] a decision-making problem inspired by a real-world scenario that can be modelled using a representation language such as PDDL (McDermott et al. 1998). For instance, from a representation point of view, the grid could be modelled as a set of interconnected nodes denoting locations in the warehouse for objects and agents (e.g., vans, boxes, and robots), as illustrated in Figure 1. A trivial example of a goal might be to

---

[1]https://github.com/potassco/pddl-instances/tree/master/ipc-2011/domains/floor-tile-sequential-satisficing

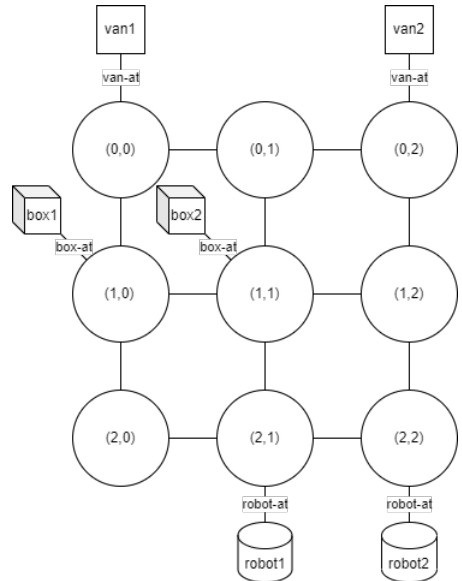

Figure 1: Warehouse planning environment.

```
LEFT  (R1,C2-1,C2-0)   │ LEFT  (R1,C2-1,C2-0)
UP    (R1,C2-0,C1-0)   │ UP    (R1,C2-0,C1-0)
PICKUP(R1,B1,C1-0)     │ UP    (R1,C1-0,C0-0)
UP    (R1,C1-0,C0-0)   │ PICKUP (R1,B1,C1-0)
LOAD  (R1,B1,C0-0,V1)  │ LOAD  (R1,B1,C0-0,V1)
```

Figure 2: Example plan outputs for the Warehouse problem.

ensure that particular objects are in specific locations (e.g., *box1* is in *van1*).

Using the above model, we can quickly find a *valid* solution to the problem using classical automated planning techniques. For instance, Figure 2 (left) shows a plan generated by the FastDownward planner (Helmert 2006) for the problem in Figure 1, where a robot moves to grid cell *(1,0)* to pick up the box before delivering it to the van at *(0,0)*.

Figure 2 (right) shows an alternative action sequence, generated using an incorrect version of the domain. Although the plan is similar to the one on the left, it is *incorrect*: the robot executes the pickup action when in grid cell *(0,0)* be-

fore loading the van. (This plan is the result of a missing precondition on the pickup action which normally ensures that the robot and object are in the same cell). While this kind of error can be trivial to debug and correct by an expert knowledge engineer, this isn't always the case for novices in languages such as PDDL. Catching modelling errors, such as incorrect logic in action preconditions and effects or missing properties in the initial state, can still be difficult due to the complexity of the knowledge that needs to be specified and the level of abstraction that is often required for ensuring the generation of tractable solutions.

In this paper, we present the Planning Domain Simulation (PDSim) (Anonymous 2020, 2021, 2022, 2023) system, a framework for visualising and simulating a range of planning problems such as classical, numerical and temporal using the Unified Planning (UP) Library (Micheli and Bit-Monnot 2022) and the Unity game engine (Unity Technologies 2022). Using the UP library of PDDL the user can define the domain knowledge and the problem formulation (e.g., planner requirements, types and objects, plus standard definitions of the domain and problem). A planner then uses this information to check that a solution exists and to generate a plan that satisfies the goal. Using the generated plan, PDSim interprets the action effects as 3D animations and graphics effects in Unity to deliver a visual representation of the world and its actions during plan execution, which can aid the user in assessing the validity of the plan during execution.

While several tools already exist to aid in the process of validating planning models—notably plan validation tools like VAL (Howey and Long 2003) and formal plan verification methods such as (Bensalem, Havelund, and Orlandini 2014; Cimatti, Micheli, and Roveri 2017; Hill, Komendantskaya, and Petrick 2020)—approaches based on visual simulation and visual feedback can also play an important role in addressing the problem of correctly modelling planning domains: visual tools can serve as powerful environments for displaying, inspecting, and simulating the planning process, which can aid in plan explainability for human users (Fox, Long, and Magazzeni 2017).

In this paper, we describe the structure, components, and features of PDSim that are responsible for providing visualisations, and illustrate how PDSim can be used to simulate planning problems. PDSim is built by extending the Unity game engine editor (Unity Technologies 2022) and can use the components offered by the engine such as a path planner, scene management, and visual scripting, among others. The system uses a backend server that is responsible for defining planning problems either using the Python UP library or PDDL managing plan generation, and problem compilation, and providing support for a wide range of modelling features, such as typing, temporal actions, and action costs.

The rest of the paper is organised as follows. First, we review work related to plan visualisation and verification. We then describe how knowledge is represented in PDSim and outline the structure of the main components of PDSim, providing examples of their use by illustrating a number of planning domains. Finally, we conclude with future work and planned additions to PDSim.

## Background and Related Work

### Automated Planning with PDDL

Automated planning is a decision-making task that involves reasoning about the sequence of actions (a plan) that achieves a set of goals (Ghallab, Nau, and Traverso 2004; Haslum et al. 2019). A planning problem $\Pi$ can be thought of as a tuple $\Pi = \langle P, A, I, G \rangle$, where $P$ is a set of properties that define a state space (including possibly a set of objects), $A$ is a set of actions, $I$ is a set of initial state properties, and $G$ is the set of goal conditions to be achieved. It is useful to think of a planning problem as a state transition system, where a state captures all the properties that are true at some point in time, and actions transition states to new states. A solution to the planning problem is a sequence of actions, called a plan, that when applied transitions the initial state $I$ to a state in which the goal conditions $G$ are true.

Automated planning has been used in a variety of applications such as robotics, video games, logistics, and natural language processing. Intuitively, planning can be thought of as a search process that enables an autonomous agent (a robot or software agent) to generate a plan to achieve its goals. In this view, plan generation may typically involve the following steps:

1. **Problem Definition**: Specifying the planning model $\Pi$ (properties, actions, initial state, and goals) that captures the operating environment of the agent.

2. **Search Space Generation**: Creating a representation of the possible states that can be achieved by applying actions from the initial state to the goal state.

3. **Search**: Applying a search algorithm that explores the state space and selects an appropriate plan that satisfies the goal.

Planning problems are composed of two parts: the domain definition which specifies the state properties and actions, and the problem definition which specifies the initial state and the goal. State properties are specified using (parameterized) predicates that can be true or false in a given state, and can capture attributes of the environment, objects, or agents. For instance, (`clear cell_0_1`) might denote that location (0,1) is empty, (`at box1 cell_1_1`) might capture the fact that box1 is at location (1,1), and (`robot-empty robot1`) might represent the idea that robot1 isn't carrying anything. Predicates specify the initial state of the planning problem and the goal conditions and are also used to describe the preconditions and effects of actions.

Actions are formalised using a schema that specifies the parameters, preconditions, and effects of each action, as in Figure 3 using the PDDL language or Figure 4 using the UP library and python. The preconditions capture the conditions that must be true in a state to perform the action, while the effects describe the state changes after an action is performed. For instance, the `load-truck` action in Figure 3 and 4 has three parameters: a package (`?p`), a truck (`?t`), and a location (`?l`). A package `?p` can be loaded onto a truck `?t` provided `?t` is at location `?l`, (`at ?t ?l`), and `?p` is at location `?l`, (`at ?p ?l`). As a result of applying the action, the package will no longer be at `?l`, (`not (at`

```
(:action load-truck
    :parameters (?p, ?t, ?l)
    :precondition (and (at ?t ?l)
                       (at ?p ?l))
    :effect (and (not (at ?p ?l))
                 (in ?p ?t))
)
```

Figure 3: PDDL action representation.

```
lt = InstantaneousAction("loadTruck",
        p=parcel, t=truck, l=location)

lt.add_precondition(at(lt.t, lt.l)
                    &
                    at(lt.p, lt.l))
lt.add_effect(at(lt.p, lt.l), False)
lt.add_effect(in(lt.p, lt.t), True)
```

Figure 4: UP python action representation.

```
(move robot1 office storage_room)
(pick_up robot1 box3 storage_room)
(move robot1 storage_room load_bay)
(load robot1 van2 box3)
```

Figure 5: Example plan for the Warehouse problem.

?p ?l)), and will be in the truck, (in ?p ?t). When an action is chosen by the planner to be part of the plan, its parameters will be replaced by objects in the planning problem (e.g., van1 for ?t, box2 for ?p, and cell_1_2 for ?l).

Domain and problem definitions are used as input to an automated planner that can reason about the changes in the world state when actions are applied, and generate a plan that achieves the goal conditions. A plan is typically a sequence of actions, as shown in Figure 5, where each row represents and action and its (grounded) parameters, where the parameters in the action schema have been replaced with objects or agents from the problem definition.

**Plan Visualisation**

PDSim (Anonymous 2023) is part of the small ecosystem of simulators for automated planning which use visual cues and animations to translate the output of a plan into a 3D or 2D environment. The closest approach to ours is Planimation (Chen et al. 2020) which uses Unity as the front-end engine to display objects and animate their position while following a given plan. Planimation defines animations using an ad hoc language (namely, an animation profile) similar to PDDL. This differs from PDSim, where animations are defined using Unity's visual scripting system.[2]

The Logic Planning Simulator (LPS) (Tapia, San Segundo, and Artieda 2015) also provides a planning simu-

lation system that represents PDDL objects with 3D models in a user-customisable environment. The approach is integrated with a SAT-based planner and a user interface that enables plan execution to be simulated while visualising updates to the world state and individual PDDL properties in the 3D environment. LPS is not based on Unity but provides the user with a simple interface for plan visualisation. Several user-specified files are also required to define 3D object meshes, the relationship between PDDL elements and 3D objects, and the specific animation effects.

vPlanSim (Roberts et al. 2021) is a similar application that also aims to provide a 3D visualization of a plan but with a number of important differences. While vPlanSim offers a simple and fast custom graphical environment for creating plan simulations with few dependencies, PDSim uses the Unity game engine to offer the user industry-standard tools for creating realistic scenarios. PDSim also provides a language-agnostic tool to set up simulations which is key for users who are not familiar with PDDL and Unity.

Several systems also exist to help users formalise planning domains and problems through user-friendly interfaces. For instance, GIPO (Simpson, Kitchin, and McCluskey 2007), ItSimple (Vaquero et al. 2007) and VIZ (Vodrázka and Chrpa 2010) use graphical illustrations of the domain and problem elements, removing the requirement of PDDL language knowledge, to help new users approach planning domain modelling for the first time. Tools such as Web Planner (Magnaguagno et al. 2017) and Planning.Domains (Muise 2016) use Gantt charts or tree-like visualisations to illustrate generated plans and the state spaces searched by a particular planning algorithm. PlanCurves (Le Bras et al. 2020) uses a novel interface based on time curves (Bach et al. 2015) to display timeline-based multiagent temporal plans distorted to illustrate the similarity between states. All of these tools attempt to assist users in understanding how a plan is generated and to help detect potential errors in the modelling process.

Simulators are also prevalent in robotics applications, and multiple systems make use of game engines to provide virtual environments, such as MORSE (Echeverria et al. 2011) or Drone Sim Lab (Ganoni and Mukundan 2017). Game engines also offer several benefits such as multiple rendering cameras, physics engines, realistic post-processing effects, and audio engines, without the need to implement these features from scratch (Ganoni and Mukundan 2017), making them desirable tools for simulation. For example, Unity has been used as a tool for data visualisation, architectural prototypes, robotics simulation (Green et al. 2020), and synthetic data generation for computer vision (James Fort and Davis 2021) and machine learning applications (Haas 2014; Craighead, Burke, and Murphy 2008). There are also interesting use cases of Unity related to AI and planning, including the Unity AI Planner,[3] an integrated planner being created by Unity as a component for developing AI solutions for videogames, and Unity's machine learning agents,[4] a so-

---

[2]https://docs.unity3d.com/Packages/com.unity.visualscripting@1.7/manual/vs-nodes-reference.html

[3]Unity AI Planner: https://docs.Unity3d.com/Packages/com.Unity.ai.planner@0.0/manual/index.html

[4]Machine Learning Agents: https://github.com/Unity-

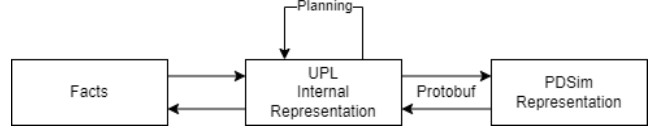

Figure 6: Representations Mappings

lution for training and displaying agents whose behaviour is driven by an external machine learning component.

## Knowledge Representation

The Planning Domain Simulation (PDSim) system is a plan and state visualizer that operates in the Unity game environment. We approach the problem of knowledge visualization by defining a planning problem using the UP library or by using the latter to parse a PDDL representation. After a plan is generated, both the plan and the problem definition are converted to a protocol buffer[5] representation that will be later mapped to Unity's game engine objects. In unity, the user defines the procedures and animations, and final visualising the plan. In this section, we discuss the underlying Planning model of Unity mappings and the representation that is used.

### Mapping planning components into Unity

Unity does not have built-in support for planning problem modelling languages but instead uses C# as a scripting language. As a result, components must be mapped into C# constructs (classes) to be represented in Unity. For a given planning domain and problem description, a set of basic constructs must be translated for plan visualisation: predicates, actions, constants, and types. Figure 6 shows the underlying general diagram of how the different knowledge representations are used and manipulated during a visualisation with PDSim. The 'facts' correspond to the high level of knowledge that the user wants to represent e.g.: a user wants to represent how a robot interacts in a warehouse environment. This high-level representation can be mapped to the 'UP Internal Representation' by using the PDDL language or the Python library. Here the representation is used to perform the search and all sorts of knowledge manipulation regarding the planning aspect. The last block corresponds to the 'PDSim Representation' that maps the planning modelling to Unity C# components using the protocol buffer representation.

Predicates define the properties of objects that can hold (or not) in a particular state. In PDSim predicates are encoded as Object Oriented Programming (OOP) classes. In particular, PDSim differentiates between Boolean, Numeric or Symbolic predicates. Boolean represent predicates that can be either true or false the animation can split in two way and the user can customise the behaviour of both values. Numeric represents a predicate that can hold a numeric value the animations can map the assignment the increase or decrease respectively. Finally the symbolic is used to map

Technologies/ml-agents

[5]Protocol Buffers: https://protobuf.dev/

animations to predicates that have a symbolic value (such as a constant). Actions are defined by their preconditions and effects. Actions in PDSim are also represented by classes that store the set of effects and all the possible objects that can be used with the action. Types are used to define a specific property for an object, in a parent-child relationship. In C#, types are represented with a tree-type structure so that if an object is of a particular type it inherits all the possible actions that the supertype has access to. For example, a *robot* can be a *physicObject* child type that inherits all the animation available to this type. Although types are not a necessary requisite for PDSim as a predicate animation can also be used to define the type of constant on the Unity side, for example, $cube(?c)$ can be mapped to an animation that can spawn a cube model or sprite and set its position in the 3D environment. Constants are used to refer to specific objects in the planning problem. In C#, and more particularly in Unity, constants represent the virtual actors in the scene. These can be 3D or 2D models and the animations that are directly applied to them.

### From the planning model to PDSim

The planning model is converted to a protobuf representation that maps to a C# model internal to PDSim representing the components presented above (predicates, actions, constants, etc.) that are used in Unity to set up the simulation. Domain entities such as actions, types, and predicates are used to set up the core Unity simulation. Similarly, problem components such as constants and the initial state are used to set up a Unity-level scene. Once these components are defined, the user can customise them using the Unity editor, for instance configuring multiple problems for the same domain, or multiple simulations for different plans.

Figure 7 shows the PDDL problem definition for the initial state described in the introduction. The `at` predicate is used to describe the position of a physical object (robots, boxes and vans), `robot-empty` represents if a robot is carrying a box or not, `van-request` represents which box is requested by a particular van, and `up`, `down`, `right`, `left` represent the connections between cells in the grid. The same PDDL representation can be visualized with 3D models in PDSim as shown in Figure 8. The PDDL `:init` block from Figure 7 can be animated in PDSim by assigning translation sequences to the physical objects and displaying them in game mode.

### From C# to Animations

The planning domain description is used to build the core elements and animations for the simulation. The types and objects define the visual aspect of the simulation in Unity: 3D models or 2D sprites. Once mapped, predicates are used to define the 2D/3D animations using the visual scripting option in Unity. This visual scripting language is used to define common transformation operations, path planning, audio emission, particle effects, etc.

For example, Figure 9 shows an animation definition for the earlier Warehouse planning problem, for a predicate that captures the movement of the robot position from the current grid to an adjacent cell. Action effects are the animated

```
(:init
    (at box1 cell_1_1)
    (at box2 cell_1_2)
    (at robot1 cell_0_0)
    (at robot2 cell_2_2)
    (robot-empty robot1)
    (robot-empty robot2)
    (at van1 cell_2_2)
    (van-request van1 box1)
    (clear cell_0_1)
    (clear cell_0_2)
    (clear cell_0_3)
    ...
    (up cell_0_1 cell_1_1)
    (down cell_1_1 cell_0_1)
    (right cell_0_2 cell_0_1)
    (left cell_0_1 cell_0_2)
...)
```

Figure 7: Initial state example.

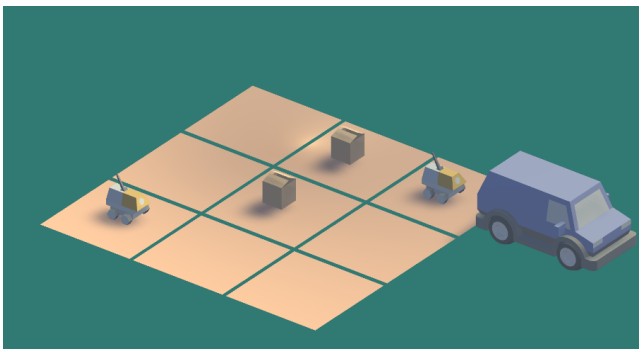

Figure 8: Initial state representation in PDSim.

components, where every predicate in the effects list that has an associated animation graph will execute an animation at simulation time.

The Algorithm 1 shows how animations are selected to be scheduled for execution. When a plan is executed PDSim look in the mapped actions definition to check the effects of an action. If one of those effects has an animation defined by the user it will start the animation loop as illustrated in Figure 10. This animation loop is based on a simple state machine where the connection between states are:

1. Default behaviour if fluents exist
2. No more fluents to animate in the queue
3. Queue has fluents to animate
4. Animation exists and is selected
5. Animation has finished
6. Queue still has animation scheduled
7. No more fluents to animate in the queue

Users can define their own behaviours in the virtual scene for every predicate they want to animate. The example in

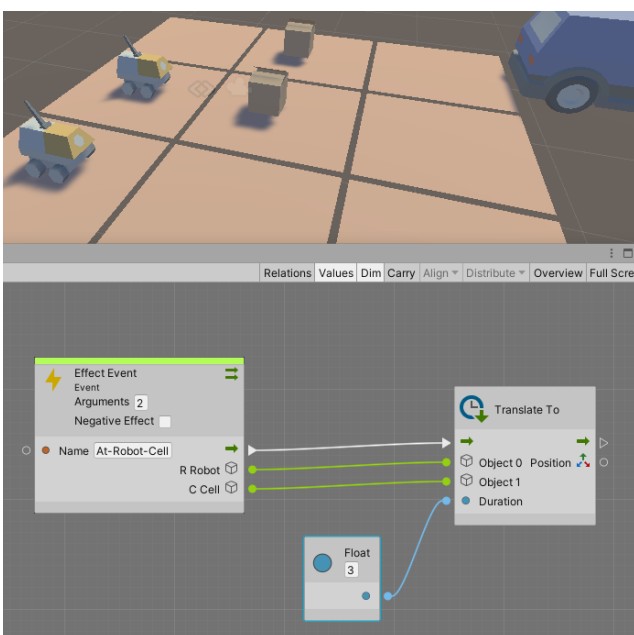

Figure 9: Example animation definition.

Figure 9 shows a simple translation animation from an object position to a target position. In particular, the example shows one of the custom animation nodes developed in PDSim to help simplify the creation of animations for new users. Every predicate in an action's effect can have one of these graphs linked to it, and every graph comes with an *EffectEvent* that is invoked during plan simulation with the corresponding objects from the Unity scene (i.e., the objects in the plan's action).

To simplify the development of new animations, and to help new users with visual scripting, a set of predefined animation nodes has been created which cover a number of useful simulation cases that frequently arise, such as:

1. **TranslateToPoint**: Move a particular object in the scene to a specific point in the world or to another object's position (using path planning or simple interpolation).

2. **TranslateToObject**: Move a particular object in the scene to a specific other object in the world (using path planning or simple interpolation).

3. **SpawnObject**: Instantiate an object (i.e., a 3D mesh) in the scene.

4. **PlayPauseParticle**: Create and either play or pause a particle effect.

5. **PlayPauseSound**: Create and either play or pause an audio effect.

6. **GetCurrentPlanactions**: Can get the current simulated actions. Multiple actions can return if the simulated plan is temporal and multiple actions are currently being executed. It also returns metadata for the action such as parameters (objects) and action duration.

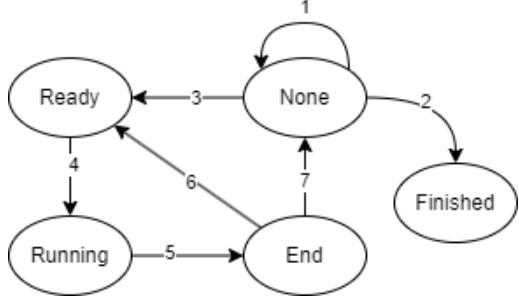

Figure 10: Fluent animation State machine

---

Algorithm 1: Animation Selection

**Input**: plan

  1: Create $animationque$.
  2: Get $actions\_map\langle name, action\rangle$
  3: Get $predicates_map\langle name, predicate\rangle$
  4: **while** plan has actions **do**
  5:    **if** temporal plan **then**
  6:      Group $action$ with same $initial\_time$
  7:    **end if**
  8:    Get $action$ from $actions\_map$
  9:    **repeat**
10:      Let $animations$=ENUMERATE $action.effects$
11:      **if** $animations\_map$ CONTAINS $animations$ **then**
12:        AnimationLoop($animations$)
13:      **end if**
14:    **until** No effects
15: **end while**

---

## System Architecture

The high-level structure of the PDSim system is shown in Figure 11. The PDSim system can be imported into Unity3D as a common asset, where the Unity editor interface is used to interact with PDSim components, such as setting the simulation scene, creating animations, or importing 3D or 2D models. PDSim also relies on a Python backend implementation, which is used to parse PDDL files and generate plans. A PDSim simulation is initialised and handled by the backend server running the Unified Planning Library (UPL[6]), which is responsible for parsing and building a Protobuf representation of the planning model and running a user-defined planner (defaulting to FastDownward) to generate a plan. UPL is a planner-agnostic framework for Python, which increases PDSim's modularity and lets users select their preferred planner implementation, separating it from the simulation stage itself which comes later in the process. We describe the major components of PDSim below.

### Front-End

Unity (Unity Technologies 2022) is a popular state-of-the-art game engine used for building 3D projects across a range

---

[6]https://github.com/aiplan4eu/unified-planning

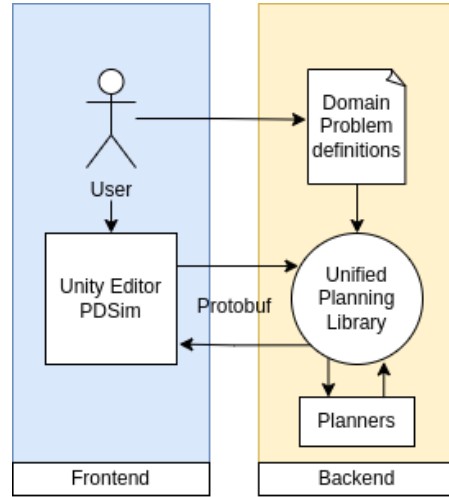

Figure 11: High-level PDSim system architecture.

of diverse applications. In PDSim, Unity provides the frontend interface and is responsible for handling all of the 2D/3D graphics and animations related to the simulation.

One of the fundamental design concepts used by Unity is the idea of *composition*, which means that an object can be *composed* of different types of objects. In particular, Unity's component system provides the capability for every object in a Unity scene to be assigned custom scripts or modules, such as a rigid body for the physics simulation, a collision volume, an audio source, etc. Every object in Unity can also be scripted using the C# language, meaning that an object can have a user-defined behaviour in the scene. For example, an object can respond to user inputs from a mouse or keyboard or can be translated, rotated and scaled, or have its colour changed, based on conditional events. Object scripting in Unity is key to the modularity of the simulation, especially for the custom representation of PDDL elements.

Scripting can also be applied to the editor window, where users interact with the engine and where it is possible to set the properties of the objects in the scene by using Unity's user interface. PDSim makes heavy use of all the features provided by Unity, such as the Visual Scripting Language used to create animations and events. As a result, users do not need to learn a new language to develop animations and animation graphs can be modified on the fly without waiting for scripts to be recompiled.

A type in PDSim is represented by a *simulation object*, a structure that shares similar information for all the objects defined in a planning problem. A simulation object is defined by two main components: models and control points. Models are used to visually represent the object type in the virtual world (e.g., block, airport, player, robot, etc.). These can be 3D meshes or 2D textured sprites that can be imported into the Unity editor. A user can add as many models as they like. A collision box that wraps all the models is automatically calculated to be used later in the simulation to detect the interaction with the user inputs and the collisions calculated by the physics engine. Control points are 3D vectors that represent particular points of interest in the object

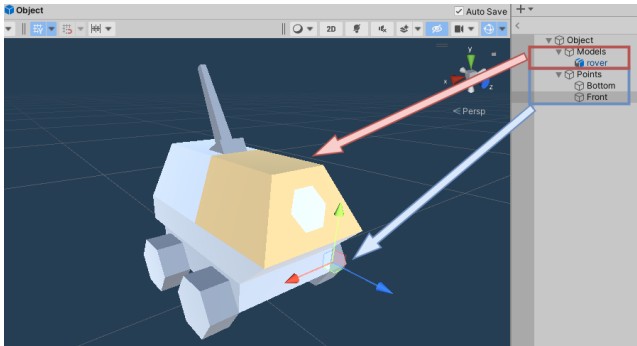
Figure 12: Simulation object example with a Robot type.

type representation (e.g., the cardinal points of an object, a point that represents the arm position of an agent, etc.).

Figure 12 shows an example of how a simulation object can be composed. The models(highlighted in red) are composed of only one mesh representing a robot rover, and the control points (highlighted in blue) are the 3D vector positions of the front and back of the robot that can be used inside the animations as an anchor point for other objects (e.g., attaching cargo on the front).

If types are specified in the domain definition, then the simulation manager creates simulation object blueprints for all the leaf types of the type tree that is built when the domain is parsed for the first time. These types are replicated for each object defined in the problem that matches the particular type, using the user configuration of simulation objects, as described above.

A simulation manager is initialised using the Protobuff data from the backend server containing the planning model and the representation of the plan. Every action effect will have an associated list of animation graphs representing the effects of an action. The simulation manager will execute the animations using the attributes in the plan representing the simulation objects involved in the simulation of that action. As the first step in every simulation, the *init* block is animated. Init represents the starting state of a planning problem and is defined by a list of fluents describing the current state of the world. These fluents are represented in the form of *fluent_name(arguments)* where the arguments are the objects that are present in the environment. The simulation manager will publish events with the corresponding fluent name and objects from the simulation scene that will be used by the visual scripting language to map which animation to execute and the graphical objects to use. The process is then repeated for every action effect in the plan.

### Back-End

PDSim's backend system is a Python server that communicates with the Unity editor and supports communication between the planning and animation components of the system. Unity tries to connect to the backend server by submitting a request using these files. The planners that can be used by PDSim are Fast-Downward (Helmert 2006), ENHSP (Scala et al. 2016), Tamer (Valentini, Micheli, and Cimatti 2020), LPG (Gerevini and Serina 2002), Aries (Bit-Monnot 2023) and Pyperplan (Alkhazraji et al. 2020). If either the parsing or planning actions fail, the interface will warn the user of the error.

PDSim's backend system wraps the functionality of the Unified Planning Library (UPL) as the main tool for manipulating and solving planning problems in PDSim. UPL is a Python library provided by the AIPlan4EU project[7] that aims to simplify the use of automated planning tools for AI application development. UPL attempts to standardize aspects of the planning process, making it accessible to users of any level of expertise. In particular, it offers a well-developed PDDL parser and a standard interface for communicating with external planners. Integration with UPL enables the PDSim system to take advantage of these features and any future updates that UPL may provide.

At the technical level, communication between PDSim's backend server and Unity is provided by the ZeroMQ networking library,[8] in particular the Python implementation package pyzmq[9] on the server side and the C# implementation netMQ[10] on the Unity side.

## Examples

PDSim has been developed and tested using the published benchmark domains from the International Planning Competition (IPC).[11] and is currently used to visualise real-world planning problems. We illustrate the capabilities of PDSim to visualise plans using as examples real-world agricultural and robotics use-case planning problems, and include a video demonstration of how to setup a simulation with the system.

### Real-world robotics

PDSim can be used to represent and visualise state changes from real-world scenarios as shown in Figure 13. The example shows a visualisation of the state changes related to sensors in a smart home. PDSim is used to play animation related for example to the robot movements between rooms or if the cupboard sensor detects it's open. This is done by connecting Unity with the Robotics Operating System (ROS) and replaying the sensor recording (ROS bags).

### Agricoltural use-case

PDSim has been used to visualise a real-world use case involving an agricultural planning problem currently being developed by the Agrotech Valley Forum [12] The problem involves a real scenario as shown in Figure 14 that has been converted into 3D models of roads and fields. There is a set of harvesters and vehicles for the transportation of grain into a silo for stocking. The vehicles can only access the fields for particular access points in the map and there is the need for a

---

[7]https://www.aiplan4eu-project.eu/

[8]https://zeromq.org/

[9]https://pypi.org/project/pyzmq/

[10]https://github.com/zeromq/netmq/

[11]https://github.com/potassco/pddl-instances

[12]https://www.ai4europe.eu/ai-community/organizations /association/agrotech-valley-forum-ev

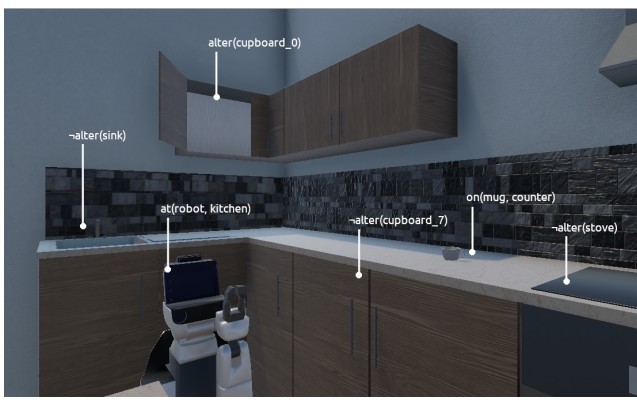

Figure 13: PDSim for real-world robotics: HSR robot (Yamamoto et al. 2019)

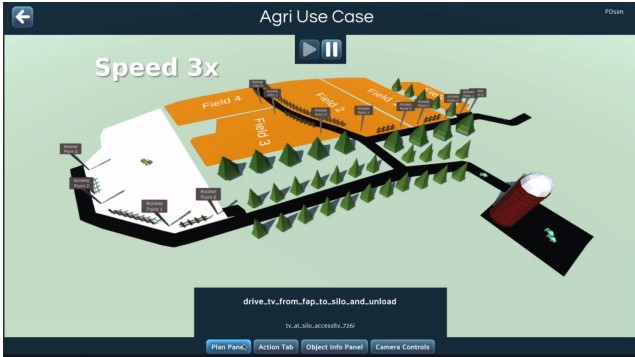

Figure 14: PDSim for an agricultural domain

planning solution to orchestrate the transporting vehicle that follows the harvesting of the fields (EV).

### Video Example

Due to the interactive nature of the system, we have created a video to demonstrate the capabilities of PDSim. The video will show how to start a new simulation from the problem definition to the final 3D animation and all the interactions with the Unity front end to customise a plan visualisation. The video is available here [13].

### Discussion

In general, PDSim offers a powerful and flexible framework for visualising planning problems using a state-of-the-art graphical engine. More specifically, PDSim aims to fill a gap in current systems that provide plan simulations, by offering users a simplified environment to develop 3D or 2D simulations, compared with current approaches that come with the overhead of learning and using an ad hoc scripting language to interact with a custom simulator (Tapia, San Segundo, and Artieda 2015; Chen et al. 2020; Roberts et al. 2021).

PDSim is designed as a support system for automated planning by providing intuitive tools to interface with a plan

---

[13]https://drive.google.com/file/d/
1AHlcYkadRa1ndJp7sxpC2VE0OTEZh0ii/view?usp=sharing

solution. Approaches like (Le Bras et al. 2020; Fox, Long, and Magazzeni 2017) also suggest that answering the question of why an action has been successfully executed or has failed, further increases the explainability of a plan. In this context, PDSim provides intuitive hints about possible errors using visual cues, by displaying an interface with the transitions of each action and how they modify the state of a particular object or agent.

It is important to reiterate, however, that PDSim is primarily aimed at planning-agnostic users like students. Within this group, as (Chen et al. 2020) indicates, there is a difference between the mental model the user has of the planning problem and the actual implementation. PDDL is often approached as a traditional programming language by beginners, rather than a knowledge definition language. With this in mind, PDSim aims to simplify the learning curve of PDDL by assisting with components that provide information about the state of planning entities in real-time.

### Conclusion and Future Work

This paper presented the structure and operation of PDSim, a simulation system for animating PDDL-based planning domains and plans. In future work, we plan to introduce a more intuitive way to create and modify the knowledge model, using the same visual scripting paradigm and thus completely removing the need to know the PDDL language syntax. This will be internally used together with an in-engine planner that the user can interact with at planning time to change object properties and replan on the fly. Given the close relationship between PDSim and Unity, it will also be possible to use applications such as extended reality (XR) to interact with the plan. Another planned direction for PDSim will also be to include extensions for visualising the current state of an agent's knowledge and beliefs to support epistemic planning, allowing visualisations to be generated from different agent perspectives. Finally, at the time of writing an evaluation is scheduled to be performed by assessing the use of PDSim in an education setting, and feedback about the overall helpfulness and usefulness of PDSim as a development aid for students learning about automated planning in an introductory AI course.

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
