# OpenReview forum: "Planning Domain Simulation: An Interactive System for Plan Visualisation"
_icaps-conference.org/ICAPS/2024/Conference — ICAPS 2024_

### Official Review · Reviewer_RWv3 · 2024-01-15

**Significance And Importance:** 2
**Soundness:** 3
**Novelty:** 2
**Clarity:** 2
**Overall Evaluation:** 1
**Confidence:** 3

**Weaknesses:**

0: Minor weaknesses requiring some work to be addressed for the paper to be accepted.

**Contributions Of The Paper:**

This paper describes a new plan visualization tool called PDSim. The aim of PDSim is to aid in the modeling, inspection and simulation of planning tasks through the use of visual feedback. PDSim provides a Unity engine interface where users can manipulate the planning task components and define animations. Moreover, by using the Unified Planning (UP) Library, PDSim supports both PDDL and programmatic representations and has the flexibility of operating with several planners and other planning tools.

**Ethical Considerations:**

(1) Not Applicable: The paper does not have any ethical considerations to address

**Nomination For Best Paper:**

No

**Questions For Authors:**

Q1. Please address my concerns regarding significance(see weaknesses)..

Q2. Are there more differences between PDSim and Planimation beyond the language used to specify animations?

**Reproducibility:**

0: N/A - nothing to reproduce.

**Strengths Of The Paper:**

Well-developed tools are huge engineering endeavours that should be valued by the community.

I expect that PDSim, like Planimation, will be an important contribution in educational settings to aid students in understanding the specification and resolution of planning problems.

The integration with UP should give PDSim great flexibility in its handling of the different planning processes including modelling, plan synthesis and validation.

**Weaknesses Of The Paper:**

Significance: While I can understand the appeal of visualisation tools for educational purposes, its usefulness in other settings is not so clear to me. I am sceptical that such tools are really helpful to an expert trying to model a domain as the paper suggests. My reasoning is that defining animations with enough fidelity to convey informative feedback that can be used for inspecting and debugging domains is as hard if not harder than defining the domain itself. In that regard, visualisation seems to require extra effort rather than make things easier, since it essentially requires modelling twice the domain.

Novelty: The discussion of the differences with respect to Planimation is insufficient. From what is written, it seems that they only differ in the language used to specify animations which I don’t find enough to motivate a whole new tool.

Presentation: The presentation of the paper is largely responsible for my negative score. My main complaint is that I don’t think that a person interested in the tool would get much out of reading this paper. This is mainly because there are too many details about the engineering of the tool. I think it would have been better to trim the contents down to fit a short paper that focuses on the usability and functionalities of PDSim and showcases these through examples.

More detailed comments about the presentation:

- Figures 1 and 2 and the example that accompanies them seem like a lot of wasted space to illustrate something fairly trivial.

- The paper includes 4 anonymised references to PDSim which raises the question of what is novel in this paper with respect to these previous publications. If these references are from un-indexed venues, maybe it was better to omit them.

- I didn’t see the point of describing engineering details such as the mapping from the UP internal representation to the C# representation used by Unity or the animation scheduling of Algorithm 1. Rather, I believe more intuitive descriptions such as the ones used starting line 321 or line 443 make more sense for this kind of paper.

- The example plan in Figure 5 seems unnecessary given that we already have example plans in Figure 2.

- The Examples section felt quite underwhelming as very little was said about the two use cases.


Evaluation: The abstract mentions a user evaluation that is not present in the paper. I think this evaluation would have been a great addition to this paper to demonstrate the advantages of using PDSim.

Lastly, I am not sure if ICAPS is the right venue for this kind of system demo papers. Reading the call for papers, it is unclear to me if this paper fits in the Application topic and I couldn’t find a demo track either. For now, my score does not consider this point and is only based on the above strengths and weaknesses.

Minor: Grammatical errors in sentences starting in lines 290 and 293.

POST-REBUTTAL
Thank you for answering my questions. Your answer to my and skRt's Q2 clarified what PDSim is capable of and how it differs from existings tools. I would recommend to include a comprehensive list of PDSim's functionalities in the manuscript and to improve the discussion of the related work to better highlight the differences with existing simulation/visualisation tools.

---

> ### Author Rebuttal · Authors · 2024-01-27
>
> Q1: Early versions of the system have already been successfully used for research work.
> https://openreview.net/forum?id=9ShFZ6a8LpI
>
> There have also been past papers appearing at ICAPS for other visualization systems (notably vPlanSim and itSIMPLE) that are compared with PDsim in the current paper, e.g.:
>
> https://ojs.aaai.org/index.php/ICAPS/article/view/15995
>
> Vaquero et al. 2007. itSIMPLE2.0: An Integrated Tool for Designing Planning Domains. In Proceedings of ICAPS, 336–343
>
> All of these papers appeared as past ICAPS Application Track papers.
>
> The user evaluation in the abstract is a typo which will be corrected. It is explained in the future work section where it states that a user evaluation is scheduled to be conducted at the time of writing the paper.
> Numerous examples showcasing the engine's functionality are available in past papers, complemented by a demonstration video. The project's GitHub repository features a dedicated demo section and a comprehensive wiki containing various examples of animations. The provided documentation outlines how to quickly set up a new visualisation.
> Importantly, the examples highlighted in the paper represent real-world domains, as opposed to the IPC domains utilized for development purposes in previous papers. This shift towards presenting practical use cases, particularly in the context of robotics presented in the paper, aims to strengthen the case for PDSim and addresses common feedback received during the development and publishing phases, to emphasise the system's relevance and applicability in real-world settings.
>
> Q2: There are several notable distinctions. In PDSim, users can inspect the current state of objects and the overall world. Furthermore, they can adjust the visualization speed, facilitating the identification of any domain definition errors that may occur during plan execution. As for the reply for skRt Q2, PDSim supports numerical and temporal problems and offers the possibility to interface with ROS, to visualise and inspect plans for robotics domains.
> Additionally, planimation only offers 2D representation to the best of our understanding whereas in PDSim the user can choose between 3D or 2D (particularly useful for robotics domains that need to represent real-world environments and not simple sprites)
> Finally, thanks to the interface with Unified Planning, PDSim supports a wide range of planners, plus it can support custom planners.

---

### Official Review · Reviewer_iiK7 · 2024-01-20

**Significance And Importance:** 2
**Soundness:** 3
**Novelty:** 2
**Clarity:** 3
**Overall Evaluation:** 2
**Confidence:** 5

**Weaknesses:**

1: Minor weaknesses that are easily fixable.

**Contributions Of The Paper:**

The authors present a simulator, PDSim, for visualizing planning problems (including classical, temporal and numerical ones) and simulating the execution of their solutions in both 2D or 3D environments by allowing them to display, inspect and simulate the planning process they can better debug their planning domains & even problems as well as gauge the correctness and the quality of plans.

It adds to the important body of work on tooling for domain modellers, and can be an asset in particular, in educational settings. Moreover, its integration into the Unified Planning framework is a definite plus that makes it more accessible.

**Ethical Considerations:**

(1) Not Applicable: The paper does not have any ethical considerations to address

**Nomination For Best Paper:**

No

**Questions For Authors:**

In the "Conclusion and Future Work" section, you state "As future work, we plan to introduce a more intuitive way to create and modify the knowledge model, using the same visual scripting paradigm, and thus completely removing the need to know the PDDL language syntax.".

Given that the main users that you specify are novice users in an educational setting and one of the stated goals is to help them learn about planning and PDDL so that they can model domains better and find errors in their existing models, this "feature" would seem to be counterproductive. Could you please explain the motivation for this?


POST-REBUTTAL
Thank you for your answer. I would recommend that you amend the section in the manuscript to clearly reflect PDSim's role in helping users learn PDDL.

**Reproducibility:**

5: Code and domains (whichever apply) are already publicly available

**Strengths Of The Paper:**

The use of an industry standard tool (the Unity game engine) is a good choice and may indeed make it more accessible to students within the robotics field in particular as they are more likely to already be familiar with it. The use of PDSim on the example problems from the IPC is another plus.

As for the paper, and given its main target audience is students who are new to planning and learning PDDL, it is written in a way that makes it accessible to them, covering the basics of automated planning and including examples.The paper also includes a good discussion of related work.  A video is available demonstrating the use and capabilities of PDSim.

**Weaknesses Of The Paper:**

- In Section Introduction, line 105, the stated origanisation of the paper is incorrect. First, you give an overview of Automated Planning with PDDL. Moreover, while you review related work on visualization, you do not do so for verififcation in that section but rather have briefly covered it in the introduction. If you rather mean that by visualizing the plan execution, verification of the domains is enabled, then perhaps update the subsection heading to "Plan Visualisation & Verification".
- Line 111: extensions is a better technical term for additions.
- Lines 145-155: a different typography is used for cell(x,y) than previously in introduction.
- Line 275, 278 & 283: wrong open quote symbol.
- Line 307: "not a necessary requisite for PDSim"--> "not necessary for PDSim".
- Line 348: etc. should only be used when the readers know precisely which terms complete the list (e.g. Monday, Tuesday, etc.).
- Line 356: Algrith 1 shows (no "The" Algorithm 1...).
- Line 397: GetCurrentPlanactions... is the 'a' for actions being in small case a typo?
- Unified Planning Library is first mentioned in the abstract and the acronym is already introduced in Section "System Architecture"... no need to reintroduce it again in Section Backend.
- The quality of figures 6,10 and 11 needs to be improved (they are pixelated).
- All the figure captions could be more informative and specific.
- The space between footnotes 12 and 13 and the words they refer to should be removed.

---

> ### Author Rebuttal · Authors · 2024-01-27
>
> Q1: Learning PDDL can be challenging for students who are accustomed to imperative languages commonly taught in standard programming courses. In the context of future work, the intention is not necessarily to eliminate PDDL but to introduce a graphical approach to represent knowledge in PDSim as an additional option. This graphical representation would be capable of being translated into PDDL, ensuring that the PDDL representation remains accessible to users at all times.

---

### Official Review · Reviewer_skRt · 2024-01-23

**Significance And Importance:** 2
**Soundness:** 4
**Novelty:** 2
**Clarity:** 4
**Overall Evaluation:** 2
**Confidence:** 4

**Weaknesses:**

0: Minor weaknesses requiring some work to be addressed for the paper to be accepted.

**Contributions Of The Paper:**

This paper introduces PDSim (Planning Domain Simulation), a system designed for visualizing plans with a dual emphasis on (1) representing and (2) manipulating domain knowledge. To address the first aspect, PDSim utilizes either PDDL or the Unified Planning Library Python. For the second aspect, it creates an interface that enables PDSim users to interact with the Unity game engine, connecting to a set of existing planners. PDSim proves versatile for various planning-related tasks, and user evaluations affirm its efficacy as a practical tool.

**Ethical Considerations:**

(1) Not Applicable: The paper does not have any ethical considerations to address

**Nomination For Best Paper:**

No

**Questions For Authors:**

In what ways does this paper differentiate itself from the preceding four? Does it essentially provide a summary of these four papers?

Moreover, how fundamentally distinct is PDSim from other existing competitors, offering a persuasive case that it constitutes a significant addition to our range of choices?

**Reproducibility:**

3: Authors describe the implementation and domains in sufficient detail.

**Strengths Of The Paper:**

The paper is well-structured and easily comprehensible. It maintains a clear theme, providing a straightforward explanation of its content.

In addition, the paper conducts a comparative analysis between PDSim and other prominent planning simulation systems. Technical intricacies are explored, such as the integration of C# components into the Unity environment.

**Weaknesses Of The Paper:**

The authors could provide more clarity on how their current system distinguishes itself from existing ones, which would help in understanding the rationale behind introducing another simulator. The paper seems to lean towards being a system description, and in fact it dedicates in introducing a new planning visualization tool.

Additionally, the mention of four previous papers on PDSim by the authors between 2020 and 2023 raises curiosity about the specific need for this particular addition. Although the anonymity of these papers makes direct verification challenging, it is suggested that, in any case upon an acceptance of the paper, the authors consider providing a more detailed justification for the inclusion of this new paper. This could involve highlighting how it builds upon or diverges from their previous contributions, thus enhancing the overall impact and relevance of the paper in the scholarly context.

---

> ### Author Rebuttal · Authors · 2024-01-27
>
> Q1: PDSim has undergone continuous development over the years and has consistently been featured in the KEPS workshop to introduce new additions and gauge interest within the planning community. Recently, external funding was pivotal for the project, enabling a more focused approach and bringing PDSim to a level of maturity that we believe now makes it a suitable candidate for presentation at the main ICAPS conference. Regarding technical enhancements, PDSim has evolved by incorporating additional in-engine tools to streamline plan visualization creation. The development process prioritized a robust backend for seamless integration with various planners and the transition from using C# for script animation to a more user-friendly visual scripting language. Furthermore, PDSim now offers the capability to connect with ROS for applications in robotics domains.
> This paper aims to present PDSim not as a prototype experiment, as often seen in workshop venues, but as a mature system. As such, the paper provides a detailed description of its structure and components. The system is positioned as actively usable for teaching, research (see robotics example), or industry applications (see agricultural domain example).
>
> Q2: Unlike other visualisation systems that offer a limited selection of planners, PDSim supports various planners such as fastdownward, tamer, and lpg to name a few, supporting numerical and temporal planning. This distinguishes PDSim from existing visualization systems. Additionally, PDSim stands out by offering compatibility with ROS for robotics a feature that, to the best of our knowledge, is not supported by any other visualization system. PDSim presents a straightforward but advanced animation definition pipeline, as demonstrated in the paper's accompanying demo video, in contrast to other visualization systems. Notably, tools like Planimation and vPlanSim differ from PDSim. Planimation relies on an Animation Profile (a definition language similar to PDDL, while vPlanSim necessitates scripting a Python routine. The reviewer must know that the animation creation visual language in PDSim is a scripting language, not a modelling language (see Planimation). A notable example and feature of PDSim is its ability to interact with the plan during runtime, allowing users to interrupt plan execution and call the planner again within PDSim to utilize the new plan. For users seeking more in-depth modifications to the visualisation, PDSim provides a C# API.

---

### Meta-Review · Area_Chair_jheB · 2024-02-01

**Recommendation:** Accept (Oral)
**Confidence:** 4

**Metareview:**

While the reviews were initially split across the borderline, the one negative reviewer's concerns were alleviated by author rebuttal and the discussion. So the overall decision is to accept.

Regarding poster vs oral: On the one hand, the work seems relevant to a smallish subset of the community. On the other hand, work on modelling tools has certainly been underrepresented in the past, and it can be useful to expose the wider community to this. Weighing the latter higher than the former, I recommend oral presentation. Though this is of course up to the discretion of the chairs given program constraints.

**Ethical Considerations:**

(4) Good: The paper adequately addresses most, but not all, of the applicable ethical considerations